A robust semi-supervised NMF model for single cell RNA-seq data

Wu Peng
An Mo
Zou Hai-Ren
Zhong Cai-Ying
Wang Wei
Wu Chang-Peng changpeng_wu@163.com
Department of Neurosurgery, The People’s Hospital of Longhua District , Shenzhen , Guangdong Province , China
Karakülah Gökhan
Electronic publication date: 2020 Oct 16
Publication date: 2020
Volume: 8
Electronic Location ID: e10091
Received 2020 Apr 27; Accepted 2020 Sep 13
Copyright: ©2020 Wu et al.
Copyright year: 2020
Copyright holder: Wu et al.
License: This is an open access article distributed under the terms of the Creative Commons Attribution License, which permits unrestricted use, distribution, reproduction and adaptation in any medium and for any purpose provided that it is properly attributed. For attribution, the original author(s), title, publication source (PeerJ) and either DOI or URL of the article must be cited.
License URL: https://creativecommons.org/licenses/by/4.0/

Keywords: Semi-supervised, NMF model, Single cell RNA-seq

Funding: The authors received no funding for this work.

==============================
Background

Single-cell RNA-sequencing (scRNA-seq) technology is a powerful tool to study organism from a single cell perspective and explore the heterogeneity between cells. Clustering is a fundamental step in scRNA-seq data analysis and it is the key to understand cell function and constitutes the basis of other advanced analysis. Nonnegative Matrix Factorization (NMF) has been widely used in clustering analysis of transcriptome data and achieved good performance. However, the existing NMF model is unsupervised and ignores known gene functions in the process of clustering. Knowledges of cell markers genes (genes that only express in specific cells) in human and model organisms have been accumulated a lot, such as the Molecular Signatures Database (MSigDB), which can be used as prior information in the clustering analysis of scRNA-seq data. Because the same kind of cells is likely to have similar biological functions and specific gene expression patterns, the marker genes of cells can be utilized as prior knowledge in the clustering analysis.

Methods

We propose a robust and semi-supervised NMF (rssNMF) model, which introduces a new variable to absorb noises of data and incorporates marker genes as prior information into a graph regularization term. We use rssNMF to solve the clustering problem of scRNA-seq data.

Results

Twelve scRNA-seq datasets with true labels are used to test the model performance and the results illustrate that our model outperforms original NMF and other common methods such as KMeans and Hierarchical Clustering. Biological significance analysis shows that rssNMF can identify key subclasses and latent biological processes. To our knowledge, this study is the first method that incorporates prior knowledge into the clustering analysis of scRNA-seq data.

Introduction

Single cell RNA-seq (scRNA-seq) is a powerful tool enabling the transcriptional profiles at cellular resolution, comparing with “bulk” RNA-seq which can only measure the average gene expression among a group of cells. Compared with traditional high density microarray in single-cell transcriptomes, scRNA-seq has an obvious advantage that it can profile all transcripts in one cell rather than only detecting known genes by complement sequence probes. Besides, by scRNA-seq we can assess the heterogeneity between cells and identify the hidden biological process, such as embryonic development and the origin of cancer cells (Tirosh et al., 2016; Zeisel et al., 2015).

Downstream analysis is the most important step in the workflow of scRNA-seq analysis and necessary for solving specific biological question. Clustering plays a fundamental and important role in many downstream analysis methods since it has a substantial impact on the outcome. There are considerable clustering algorithms for scRNA-seq and most of them can be applied to any type of data, such as KMeans and Hierarchical Clustering. However, most of these methods are unsupervised and does not consider known biological knowledge such as cell marker genes. In this paper, we proposed a semi-supervised NMF model for the clustering analysis of scRNA-seq data, which incorporates cell marker information and significantly improves the accuracy of clustering analysis.

Related work

Nonnegative matrix factorization (NMF) is an effective method for unsupervised clustering analysis of gene expression data. Given a nonnegative matrix X of size m × n, NMF aims to find two non-negative matrices W and H such that: X≈WH

Where W ∈ Rm×k is a basis matrix, H ∈ Rk×n is a coefficient matrix. The solution to this problem can be obtained by solving the following optimization problem: minW,HOW,H=minW,H12||X−WH||F2

||.||F means Frobenius norm and W and H satisfy nonnegative constrains. Since many practical problems in data mining, pattern recognition and machine learning require non-negativity constraints and appropriate low dimensional representation of original data, NMF has been successfully applied to these fields, which obtains the parts-based representation as well as enhancing the interpretability of the data (Berry et al., 2007; Chagoyen et al., 2006; Lee & Seung, 1999). In bioinformatics, it has been used to extract meaningful features that belongs to different cell types from microarray and scRNA-seq data and identify mRNA isoforms (Brunet et al., 2004; Shao & Hofer, 2017; Ye & Li, 2016).

Different variants of NMF has been put forward such as sparse NMF (SNMF) and discriminant NMF (DNMF) for microarray and RNA-seq data (Jia et al., 2015; Kim & Park, 2007). SNMF introduces a regularization term on W or H to control the degree of sparsity and generate sparser representation. DNMF incorporates Fisher’s discriminant criterion in the coefficient matrix by maximizing the distance among any samples from different classes meanwhile minimizing the dispersion between any pair of samples in the same class. The DNMF requires discriminant information to construct the objective function and has been applied in various scenes such as face recognition and facial expression recognition. Yet in most cases of scRNA-seq, we cannot know the exact class information. Moreover, none of these methods consider the technical factors including amplification, library size differences and dropouts (Buettner et al., 2015; Kharchenko, Silberstein & Scadden, 2014). In addition to the technical factors, scRNA-seq data exhibit high cell-to-cell variability in gene expression, which impeding the analysis. Overall, the technical effects and inherent variability in scRNA-seq introduce substantial noise and may corrupt the analysis of underlying biological process.

In this paper, we propose a robust semi-supervised NMF model that is robust to noises and uses marker information of cells as prior knowledge. We already have prior knowledge about the marker genes of different cells (such as MSigDB database; Liberzon et al., 2011), for instance, metastatic melanoma is a mixture of tumor cells and a variety of normal cell including T-cells, B-cells, macrophages and NK cells; thus, we can know the prior marker genes of different groups. We will show that incorporating such information into the NMF model, the clustering accuracy can be increased significantly.

Materials & Methods

Original NMF algorithm

The original NMF algorithm was introduced by Lee and Seung and the objective function is non-increasing if follows the multiplicative update rules (Lee & Seung, 2001): Wij⟵WijXHTijWHHTijHij⟵HijWTXijWTWHij

The author proved that by repeating iteration of the update rules is guaranteed to converge to a locally optimal matrix factorization. By using the non-negative constrains, NMF can learn a low dimensional representation of the original data.

Roubst NMF algorithm

Robust NMF (rNMF) first proposed by Kong and Ding and was designed to handle outliers and noises that original NMF fail to cope with (Kong, Ding & Huang, 2011). The main difference of rNMF is using L2,1 norm loss function instead of Frobenius norm which is defined as follow: ||X−WH||2,1= ∑j=1n∑h=1mX−WHhj2= ∑j=1n||xj−Whj||

In this formulation, the error of each data point is ||xj − Whj|| rather than squared, thus the errors caused by outliers and noises do not dominate the objective function compared with the squared one. Different from the revision on the objective function, Zhang proposed a method which introduce an error matrix S ∈ ℝn×m to capture the errors and handle the extreme data points (Zhang et al., 2011): minW,H,S||X−WH−S||F2s.t.W≥0,H≥0,||S||0≤υ

Where ν is the parameter that specifies the maximum number of nonzero elements in S.

Robust semi-supervised NMF algorithm

Motivated by robust NMF formulation, we adopt similar method to deal with the noises in scRNA-seq data under the assumption that the data matrix may be corrupted by noises and the noises are sparse (we use rNMF to represent this model below). In addition, we incorporate the prior information into a graph regularization term to maintain the intrinsic geometrical and discriminating structure of the data in the parts-based representation: minW,H,S||X−WH−S||F2+α|S|1+βTrHLHTs.t.W≥0,H≥0

Where Tr(.) denotes the trace of a matrix and α, β is the regularization parameters. Since the 𝔩0 − norm is difficult to solve, we replace the 𝔩0 − norm constraint with a 𝔩1 − norm regularizer, which is a common technique for sparse solution. The second graph regularization term maintains the local geometrical structure of the original data matrix X in the low dimensional representation, i.e., the matrix H. Briefly, if two data points xi and xj are close to each other in matrix X, then hi and hj, points in the low dimensional representation matrix H, are also close to each other. This property is usually referred to as local invariance assumption (Belkin & Niyogi, 2002; Cai, Wang & He, 2009) and has been applied in the development of dimensionality reduction algorithms and semi-supervised learning algorithms (Belkin & Niyogi, 2002; Zhu & Lafferty, 2005).

Chung et al. demonstrate that the local geometrical structure can be modeled by a nearest neighbor graph on a scatter of data points (Chung & Graham, 1997). For each data point xi in X, we need to find its neighbors and put edges between xi and its neighbors. In our scRNA-seq problem, we suppose that all the cells are the nodes in a graph, but different group of cells have different weighting. We use marker genes of cell group to construct the weight matrix Q, which can be defined by various ways. In this paper, we chose the heat kernel weighting and the weighting between xi and xj are: Qij=e−||xi−xj||2σ

where σ is parameter that controls the weighting and we set σ = 1 in the experiment. Note that only the rows (genes in rows and samples in columns for matrix X) that we select as cell markers of X will be used to compute the weighting. As shown in Fig. 1, the expression matrix X includes five groups and each group has its own highly-expressed genes (yellow rectangular block). The rows which are the markers of different groups are selected to construct the weight matrix.

Figure 1 (A-B) An illustration of how to construct the weight matrix Q (C).

(A) The heatmap is an ideal simulated gene expression matrix X; (B) the heatmap is part of the matrix X that only selects rows of marker genes.

We use Euclidean distance to measure the dissimilarity in the low dimensional representation of X: Dhi,hj=||hi−hj||2

then the local geometrical structure preserving criterion can be written as: ℜ= ∑i,j=1n||hi−hj||2Qij= ∑i=1nhiThiDii− ∑i,j=1nhiThjQij=TrHDHT−TrHQHT=TrHLHT

Where D is a diagonal matrix and Dii = ∑jQij. L is called graph Laplacian and L = D − Q. We can see that by minimizing ℜ, if data point xi and xj are close (Qij is big), hi and hj will also be close to each other, so the distance relation between points in original data matrix X can be preserved in low dimensional matrix H. This technique was first designed by Cai et al. and introduced to solve the embedded structure problem, which assume that the data is usually sampled from a low dimensional manifold embedded in a high dimensional ambient space (Cai et al., 2011). We show that by encoding the geometrical information in the NMF model, the clustering accuracy can be greatly improved.

Optimization algorithm for robust semi-supervised NMF

In this section, we derive the iterative multiplicative update rules for the robust semi-supervised NMF based on the coordinate descent method (Sha et al., 2007). The objective function is not convex for W, H and S together, so it is unrealistic to expect an algorithm to find the global minima. It is natural to optimize W, H and S separately since the objective function is convex while holding the other two variables as constant.

The objective function of rssNMF can be rewritten as: O=||X−WH−S||F2+α|S|1+βTrHLHT=TrX−WH−SX−WH−ST+βTrHLHT+α|S|1=TrXXT−2TrXHTWT−2TrXST+2TrWHST+TrWHHTWT+TrSST+βTrHLHT+α|S|1

After introducing the Lagrange multiplier ΨandΦ for the constrains, the Lagrange function 𝔏 is stated as L=TrXXT−2TrXHTWT−2TrXST+2TrWHST+TrWHHTWT+TrSST+βTrHLHT+α|S|1+TrΨW+TrΦH

and the partial derivatives of 𝔏 with respect to WandH are: ∂L∂W=−2XHT+2SHT+2WHHT+Ψ∂L∂H=−2WTX+2WTS+2WTWH+2βHD−2βHQ+Φ

By KKT conditions ψhiwhi = 0andϕijhij = 0, we can obtain: −XHTijwij+SHTijwij+WHHTijwij=0−WTXijhij+WTSijhij+WTWHijhij+βHDijhij−βHQijhij=0

Then the updating rules for W and H are: (1) Wij⟵WijXHTijWHHT+SHTij

(2) Hij⟵HijWTX+βHQijWTWH+WTS+βHDij

For fix W and H, we update S via the soft-thresholding operator (Hale, Yin & Zhang, 2008) and the optimization problem for S is: minS||X−WH−S||F2+α|S|1

Theorem 1: Define the soft-thresholding operator as below: Tυz=z−υ,ifz>υz+υ,ifz<−υ0,otherwise

Where z ∈ ℝ and υ > 0 and this operator can also be applied to vectors or matrices by element-wise operation. For the following 𝔩1 − norm problem: minv12||x−v||F2+α|v|1

the unique solution of v is given by Tα(x) (Hale, Yin & Zhang, 2008). Similarly, we can get the update rule for S: (3) S⟵Tα2X−WH

Based on above analysis, our algorithm for solving rssNMF is presented in Algorithm 1.

The proof of convergence analysis of our algorithm essentially follows the idea of graph NMF and ensures that the objective function of rssNMF is nonincreasing under the updating rules in Eqs. (1), (2) and (3) and could converge to a stationary point (Cai et al., 2010). Since the idea is similar to the method in the paper (Cai et al., 2010) so we did not prove it here.

Results

Dataset

Cell markers, usually referring to surface molecules on cell membrane, are different in different kinds of cells. Surface molecules that appear only on a particular type of cell are called cell markers and often used for cell type identification. For example, cancer cell specific markers often appear on the surface of cancer cells and are utilized as targets for anticancer drugs. Cell markers have been a hotspot in molecular biology study and researchers have accumulated lots of marker data for different kinds of cells, such as MSigDB (Liberzon et al., 2011). Here, we extend the concept of cell markers. We use ”cell marker genes” to represent those genes that can distinguish different types of cell, which are usually only expressed in specific type of cells or relatively highly expressed, such as differentially expressed genes. In single-cell RNA-sequencing, we already know where the sequencing samples come from and then we can have a modest degree of prior knowledge. That is, we have known what kinds of cells are in the experimental samples although there are still many kinds of cells remaining unknown. For instance, when the cancer tissue was sequenced, the samples contained a large number of normal cells in addition to cancer cells, such as lymphocytes, myeloid populations and cancer-associated fibroblasts. Taking melanoma as an example, melanoma is often known as malignant melanoma, which is a kind of cancer that stems from the pigment-containing cells known as melanocytes. Melanoma is a mixture of tumor cells and a variety of normal cells including T-cells, B-cells, macrophages and NK cells. This feature has been found in many studies, thus we can know the prior marker genes of different groups of cells in advance. We will show that incorporating such information into the NMF model, the clustering accuracy can be increased significantly.

Twelve scRNA-seq datasets are used in the experiment. All the datasets are normalized expression level (FPKM, RPKM or CPM) or counts and number of cells range from 56 to 3005 (Table 1). Almost allthe datasets except dataset “Zeisel” have true labels since they are collected from different time points during embryonic development and we provide the accession number in Table 1. The labels of dataset “Zeisel” comes from computation and can be seen as a silver dataset.

Table 1 Published ten scRNA-seq datasets used to test rssNMF model.

All the datasets are scRNA-seq data of human or mouse embryos.

Dataset	Units	GSE/ArrayExpress Number	Number of cells	Species	Number of Clusters	
Biase	FPKM	GSE57249	56	Mouse	5	
Goolam	CPM	E-MTAB-3321	124	Mouse	5	
Yan	RPKM	GSE36552	124	Human	9	
Shin	RPKM	GSE71485	256	Mouse	10	
Deng	RPKM	GSE45719	259	Mouse	10	
Leng	Normalized counts	GSE64016	460	Human	4	
Kowalczyk	TPM	GSE59114	564	Mouse	8	
Camp	FPKM	GSE75140	734	Human	9	
Chu_1	TPM	GSE75748	758	Human	6	
Chu_2	TPM	GSE75748	1,018	Human	7	
Tasic	RPKM	GSE71585	71,585	Mouse	7	
Zeisel	Counts	GSE60361	60,361	Mouse	8	
Notes.

FPKM fragments per kilobase of transcript per million mapped reads

RPKM reads per kilobase of transcript per million mapped reads

CPM counts per million mapped reads

Compared algorithms

To benchmark rssNMF, six clustering algorithms are used: NMF, rNMF, ssNMF, KMeans, Hierarchical Clustering and SC3.

NMF: F-norm formulation is adopted to cluster scRNA-seq data.

rNMF: similar to the proposed rssNMF but without the graph regularization term: minW,H,S||X−WH−S||F2+α|S|1s.t.W≥0,H≥0

ssNMF: ssNMF removes the matrix S which copes with noises and outliers but keep the graph regularization term: minW,H||X−WH||F2+βTrHLHTs.t.W≥0,H≥0

KMeans: a canonical distance-based iterative algorithm. Euclidean distance is used in the experiment.

Hierarchical clustering: we use a division-based algorithm which initially starts with all observations in a single cluster and divide samples until each cluster only contain one observation. Euclidean distance is used in the experiment and “ward.D2” method was used in R hclust() function.

SC3: consensus clustering algorithm for single-cell RNA-seq data, which is a benchmark in comparison to various clustering methods in scRNA-seq clustering analysis (Duò, Robinson & Soneson, 2018; Kiselev et al., 2017).

After factorization, we use KMeans to cluster matrix H and obtain the clustering results. The other common method is categorizing sample j directly by the largest coefficient in column hj.

To evaluate the clustering performance of robust semi-supervised NMF, we use adjusted Rand index (ARI, Supplementary File) which ranges from −1 to +1. Since cell labels are available so ARI can be calculated to measure the similarity between two data clustering results.

Prior genes selection and parameter setting

For all the datasets, we use differentially expressed genes as marker genes. For all the clusters in each dataset, we use R package “Deseq2” to identify the differentially expressed genes and select 20 differentially expressed genes with largest variance as cell marker genes for each group to construct weighting matrix. We set the same parameter value for rNMF, ssNMF and rssNMF and results are obtained by running the six methods 30 times on each dataset. For the factoring matrices W and H, we use standard normal distribution for random initialization.

Experimental results

Table 2 shows the clustering results on twelve datasets and we can see that rssNMF performs better than the most methods on almost all datasets. Here, we consider a basic circumstance that we only know one group of cells in the experiment and shows the results that one group of genes is added. All these datasets can be downloaded on NCBI Gene Expression Omnibus (GEO). The final results are evaluated by taking the average value of ARI over 30 runs. Compared with SC3, rssNMF gets better performance over seven datasets. We notice that the overall ARI on these seven datasets is relatively low (smaller than 0.4) which indicates that the cells hard to discriminate. Under such circumstance, the prior information can help improve the clustering accuracy than purely unsupervised clustering algorithm.

Table 2 Benchmarking of rssNMF against other clustering method.

All the algorithms were applied 50 times to each dataset. Parameter α for rNMF and rssNMF: 2. Parameter β for rssNMF: 2. Prior information: for each dataset, we randomly select one cluster and use 20 marker genes of the selected cluster to construct the weight matrix.

Dataset	KMeans	HC	NMF	SC3	rNMF	ssNMF	rssNMF	
Biase	0.712	0.761	0.774	0.844	0.806	0.796	0.862	
Goolam	0.304	0.310	0.387	0.731	0.43	0.642	0.657	
Yan	0.375	0.570	0.533	0.805	0.572	0.675	0.710	
Shin	0.167	0.217	0.282	0.366	0.282	0.327	0.370	
Deng	0.42	0.399	0.466	0.775	0.52	0.547	0.682	
Leng	0.057	0.009	0.112	0.179	0.14	0.165	0.213	
Kowalczyk	0.182	0.176	0.269	0.307	0.304	0.293	0.365	
Camp	0.232	0.225	0.274	0.327	0.3	0.297	0.305	
Chu_1	0.177	0.199	0.22	0.205	0.241	0.326	0.369	
Chu_2	0.204	0.242	0.314	0.312	0.314	0.322	0.357	
Tasic	0.51	0.284	0.705	0.822	0.711	0.791	0.790	
Zeisel	−4.97E−05	−9.36E−04	2.43E−03	−5.60E−04	2.65E−03	0.007	0.014	

Our rssNMF model has two critical parameters: regularization parameter α and β. Figure 2 and Fig. S1 shows the how the average performance of rssNMF varies with the parameters α and β. As we can see in the Fig. 2, the performance of rssNMF is quite stable with respect to the parameter α on all four datasets. If α∕2 > maxij(X − WH)ij, all the elements in S will be zero, thus when the parameter α is large enough, rNMF will be equivalent to the original NMF.

Figure 2 Performance of rssNMF versus parameter.

The rssNMF is stable with respect to the parameter—and achieve good performance varies from 2 to 32.

The parameter β controls the weights of the graph regularization term and we test different weights range from 2 to 20,000 (Fig. S1). The performances are stable when β is smaller than 200 in the four datasets. When β is too large, clustering accuracy decreases a lot for the graph regularization term dominates the whole formulation and covers the main information.

To investigate how the prior information influences the clustering accuracy, we test the performance of rssNMF with respect to different number of group information. For each group in one dataset, 20 marker genes are selected as prior information to compute the weighting matrix. Figure S2 shows how the ARI changes as the group marker information increases (20 markers genes of one group, 40 markers gene for two groups and so on). We can see that as the number of group information increases, the performance fluctuates but clustering accuracy is higher than only adding one group information.

To evaluate the clustering stability, we calculate the consensus matrix for NMF, rNMF and rssNMF over 30 runs. A consensus matrix M is a n × n matrix that stores, for each pair of samples, the proportion of clustering runs in which two samples are clustered together. A consensus matrix can be obtained by taking the average over the connectivity matrices for all runs. The connectivity matrix C is also an n × n matrix and defined based on a single run: Cij=1,iandjare in the same clulster0,iandjare not in the same cluster

So, if sample i and j are in the same cluster for 30 runs, then Mij will equal to 1, corresponding to red (0 corresponds to blue). Figure 3 present the factorizing matrices W, H and consensus matrix taking the average over 30 runs and the consensus matrix from NMF, rNMF and rssNMF on the dataset Yan. The color bar represents different group of cells or genes. More specifically, we assign genes to different cell groups according to the gene score, which is described in section Biological Significance Analysis section. The values in each row of W an column of H have been normalized between 0 and 1. We can see that all of the three methods have a stable clustering result on most samples and only small part of samples are clustered into multiple clusters in dataset Yan. Generally, rssNMF has the highest clustering accuracy and better stability than the other two methods.

Figure 3 Factorizing matrices W (basis matrix: A, D, G), H (coefficient matrix: B, E, H) and consensus matrix (C, F, I) respectively obtained from NMF (A, B, C), rNMF (D, E, F) and rssNMF (G, H, I) for dataset Yan with 124 cells and nine clusters.

The annotation color bar denotes nine clusters. The rows annotation of W and columns of H indicate the assignment of genes and samples for clusters. The paramete α = 2 for rNMF and rssNMF and β = 2 for rssNMF.

Biological significance analysis

In this section, we choose dataset Yan for biological significance analysis and dataset Yan are human embryonic stem cells including zygote, 2-cell, 4-cell, 8-cell, morula and blastocyst. In Fig. 3, each column of matrix W defines a metagene (biological procesess or pathways) or a metasample (cell cluster) then entry wij can be regarded as the coefficient of gene I in metagene or metasample j (Brunet et al., 2004). Correspondingly, a column vector hj represents the expression contribution of sample j to k biological processes, and Hij can be seen as the contribution of metagene or metasample i in sample j. The class of samples is denoted by column annotation bar. We can see that for some samples (columns of H), it is hard to visually determine which cluster it belongs to since the depth of color is close in different clusters. Another obvious feature is that compared with NMF and rNMF, the pattern of H in rssNMF is clearer and more concentrated, which makes the classification characteristics of samples more explicit. We suppose that the addition of prior information makes the differences between different clusters more obvious, thus achieving better classification results.

For the basis matrix W, the ith row of W represents the contribution of gene i to all metagenes so the expression level of a specific gene in one cell is determined by the linear combination of its contribution to all biological processes. From Fig. S3 we can see that a gene can participate in multiple biological processes, but it is more reasonable to focus on the process with the largest value. We define a gene score to determine cluster-specific genes and the assignment of cluster-specific genes is shown in row annotation bar of matrix W. The gene score for the ith gene is: Gene_scorei=1+1log2k∑j=1kpi,jlog2pi,j,

where p(i, Ω) is the probability that the ith gene contributes to cluster Ω, i.e., pi,Ω=WiΩ∕∑j=1kWij. The gene score is a real value and ranges from [0, 1]. The higher gene score is, the more cluster-specific the corresponding gene. Simply we can think that a gene is highly-expressed if the gene has high Gene_score in a specific cluster over other clusters.

As shown in the A, D and G which are the basis matrix W from NMF, rNMF and rssNMF in the Fig. 3, we assign all the genes to a specific cluster using gene score by different colors as shown in the left color bar. For the basis matrix W, the rows indicate genes and the columns indicate clusters andthe depth of the color in the matrix indicates the size of the value in W. Deep color represents a larger value and we can assign the cluster genes visually. After classifying each gene, we need to validate the biological significance of feature genes detected in different cluster. Table S1 present the results of enrichment analysis for cluster-specific genes.

We use the KEGG database to investigate the functional genes selected for all clusters. KEGG is a database resource for understanding high-level functions and utilities of the biological system, such as the cell, the organism and the ecosystem, from molecular-level information, especially large-scale molecular datasets generated by genome sequencing and other high-throughput experimental technologies. Through functional enrichment analysis of genes in each cell cluster, we can understand what biological processes are involved in the cell cluster and further infer the role of the cluster in the biological microenvironment.

Embryonic stem cells are highly undifferentiated cells. It has developmental omnipotence and can differentiate all tissues and organs of adult animals, including germ cells. The research and utilization of ES cells is one of the core topics in the field of bioengineering. The enrichment results (Table S1) show that embryonic stem cells are in an active metabolic and proliferative state, such as Ribosome pathway in cluster 3, 5, 6, 7 and 8. Ribosomes are organelles that are responsible for making proteins and are widespread. The manufacture of ribosomes requires hundreds of cytokines that are not found in mature particles. In the absence of these factors, ribosome production will stagnate. Once ribosome production is stagnant, cell growth is terminated even under optimal growth conditions. Besides, various metabolic pathways such as TCA cycle, oxidative Phosphorylation and cholesterol metabolism also occur in multiple clusters, indicating that these clusters are in an active proliferation state. In addition, we can infer the differentiation path of stem cells from the results of enrichment. Cluster 6 contains two distinctive pathways, Cardiac muscle contraction and Retrograde endocannabinoid signaling. Both Cardiac muscle contraction and Retrograde endocannabinoid signaling are involved in nerve signal transduction and are usually active in the nervous system and cardiac conduction system. Endogenous cannabinoids (endocannabinoids) serve as retrograde messengers at synapses in various regions of the brain. Therefore, it can be inferred that cells in cluster 6 are differentiating toward the ectoderm and will form epidermis and nervous system in the future. Thermogenesis pathway appears in cluster 5, 6 and 7 and this pathway usually activates in brown adipose tissues. Brown adipose tissues originate from mesenchymal stem cells which differentiate from ectoderm. This is consistent with our view that cluster 7 belongs to ectoderm cell groups.

Discussion

Considering data noises and absorbing prior knowledge at the same time in single-cell RNA-sequencing data, the clustering accuracy is significantly improved compared with NMF, which is an unsupervised clustering method. Compared to NMF, our model has higher clustering accuracy and can discover hidden structures in the data after adding the prior information. Our method also achieves better performance in part of the datasets compared to other unsupervised clustering method such as SC3, which is seen as a benchmark in scRNA-seq clustering analysis. We notice that rssNMF gets higher accuracy upon datasets that most unsupervised methods perform bad such as datasets Leng, Kowalczyk and Zeisel. We supposed that in the situation that cells are much alike each other, semi-supervised approach can get better performance over unsupervised clustering method. In addition, our approach can be used in a wider range of areas, such as medical text information mining and biological network motif identification, provided that there are noises in the data and that the characteristics of some of the samples are known. However, our method still has some limitations. Firstly, the objective function of rssNMF is based on Frobenius norm, without considering other objective functions, such as KL divergence and L2,1 norm. Some studies have found that KL divergence can achieve higher accuracy in clustering analysis of gene expression data than Frobenius norm. Therefore, a semi-supervised NMF model based on different objective functions deserves further exploration. Second, the choice of weighting matrix in rssNMF is arbitrary and we use heat kernel weighting in our model. Other weight matrix such as 0-1 weighting, and Dot-Product weighting have not been tried. In general, the choice of weighting matrix is empirical and depends on the dataset. Third, rssNMF does not take into account the statistical dependency between latent variables. In rssNMF, the gene expression level of single cell is determined by the linear combination of potential biological processes (metagenes), and the dependence relationship between these biological processes is neglected. In fact, different biological processes are interrelated and regulated with each other. In Fig. S3, we can see that the CELL CYCLE intersects the MAPK signaling pathway, Apoptosis and Ubiquitous mediated proteolysis, and NMF cannot identify the relationship between them.

Conclusions

We present a novel robust semi-supervised NMF model called rssNMF for scRNA-seq data and the model remarkably improve clustering accuracy when the cell marker information for one group or more groups are available. Compared to NMF and other common clustering algorithms, our model has higher clustering accuracy and can discover hidden structures in the data after adding the prior information. Our model also outperforms SC3 in part of the datasets while the prior information is available. What’s more, the model is robust to noises and outliers compared with standard NMF model. Our model can also be applied into other clustering tasks as long as feature information of some samples are known, such as electronic health records, single cell methylation sequencing and proteomics.

Supplemental Information

Supplemental Information 1 R code of NMF and NMF related variants

The experimental RNA-seq data are available at GEO.

Click here for additional data file.

Supplemental Information 2 Supplementary Material

Click here for additional data file.

Additional Information and Declarations

Competing Interests

Author Contributions

Data Availability

The authors declare there are no competing interests.

Peng Wu conceived and designed the experiments, performed the experiments, analyzed the data, authored or reviewed drafts of the paper, and approved the final draft.

Mo An analyzed the data, prepared figures and/or tables, and approved the final draft.

Hai-ren Zou performed the experiments, prepared figures and/or tables, and approved the final draft.

Cai-Ying Zhong and Wei Wang analyzed the data, authored or reviewed drafts of the paper, and approved the final draft.

Chang-Peng Wu conceived and designed the experiments, analyzed the data, prepared figures and/or tables, authored or reviewed drafts of the paper, and approved the final draft.

The following information was supplied regarding data availability:

R code of NMF and NMF related variants are available as a Supplemental File. The experimental RNA-seq data are available at GEO and all the accession numbers are available in Table 1.

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
