# Peer review of "A robust semi-supervised NMF model for single cell RNA-seq data"

_PeerJ, doi:10.7717/peerj.10091_

## Round 0.1 · original submission · Major Revisions

The manuscript is borderline. For further consideration, the authors must carefully and satisfactorily implement all referee comments.

In addition to this, the referees have also recommended that the language used in the manuscript should be improved. I agree with this evaluation and I would therefore request that the manuscript be revised accordingly.

The authors also should consider the following citations which are highly relevant with the clustering methods of single cell expression data: PMID:30271584 and PMID:31646845 in their revised manuscript and compare rssNMF with at least SC3 suggested as the "optimum" clustering method by Duò et al. 2018 and Suner 2019.

Reviewer 1 ·

Basic reporting

The authors of the manuscript “A robust semi-supervised NMF model for single cell RNA-seq data” present a novel approach to incorporate knowledge of tissue-specific marker genes by modifying an existing method called robust NMF to improve single cell RNA-seq data clustering accuracy. Authors demonstrate improved performance of the new method on ten single cell RNA-seq datasets that have true labels. However the application of this method to any other dataset of unknown cellular composition is not provided. Also there is no discussion in the manuscript what are the advantages and disadvantages of semi-supervised approach over commonly used unsupervised methods. The latter is important because use of the prior knowledge in clustering can lead to overfitting.

English language used throughout:
Text in 45, 47, 139 etc., should be rewritten to correct language errors.
Revise term “Hierarch clustering” and also provide parameters that were used with this clustering

Typographical and other errors in mathematical expressions:
Line 34: twelve datasets are mentioned, however in the main text analysis of ten datasets is given.
Line 118: variable sigma is not defined
Line 155, 241: “=” sign is missing

Figures:
Text on Fig. 2 is inconsistent with caption of this figure.
Supplementary Fig. 3 is similar in contents to Fig. 3 but for another dataset, and is not referenced or explained in the manuscript text.
Fig. 3 color coding is not explained.

All the questions and comments above need to be addressed in detailed.

Experimental design

Authors claim that the clustering accuracy is significantly improved when using rssNMF compared to NMF. This is justified by numbers in Tab. 2, where rssNMF is calculated with 1 group with 20 markers. However using only 1 group is not optimal, as suggested by Supplementary Fig. 1. Authors should comment why 1 group was chosen instead of 2 or 3, etc.

On line 220 authors state “Supplementary Fig. 1 shows the how the average performance of rssNMF varies with the parameters α and

Validity of the findings

In the abstract, Results subsection, it is stated that “rssNMF can identify key subclasses and latent biological processes”. However in the Results section of the main text there is no clear explanation how to identify biological processes with rssNMF method. Therefore the validation of the clustering results with pathway enrichment analysis should be clearly identified as such.

On line 266 authors reference Supp. Fig. 3, that is supposed to be Supp. Fig. 4, to exemplify genes participation in multiple biological processes. What particular gene is exemplified and what relevant processes should the reader focus on?

All the questions and comments above need to be addressed in detailed.

Additional comments

Material and Methods section needs to be rewritten to clearly separate the existing rNMF and proposed rssNMF methodology.

Derivation notes in section “Optimization algorithm for robust semi-supervised NMF” as well as Algorithm 1 are not essential for the definition of the rssNMF methods and could be moved to the Supplementary information.

Subsection “Dataset” of section “Results” contains background information on cell marker genes, and should be incorporated in the introduction section. The description of the used datasets should be extended to include information on how the clusters and true labels (annotation) in the analyzed datasets were obtained. The supplied datasets information can be improved by adding GEO accession numbers.

The mathematical definition of Frobenius norm in subsection “Original NMF algorithm” need to be moved to Supplementary Information.

The mathematical definition of ARI in subsection “Compared algorithms” need to be moved to Supplementary Information.

On lines 267-278 authors present “gene score” however the relation of this score to the rssNMF or the biological significance analysis is not clear. Also the Fig. 3 is intended to reflect the importance of this score needs more detailed explanation.
All the questions and comments above need to be addressed in detailed.

Reviewer 2 ·

Basic reporting

The majority of the manuscript is clear, but the article can be improved with a better flow and transition sentences, and some typos need to be corrected.

Experimental design

Do not apply.

Validity of the findings

The authors need to provide the accession numbers of the single cell datasets.

Additional comments

Wu et al. presented the so-called “robust semi-supervised NMF” model to cluster single cells based on their gene expression data. The main contribution is proposing to add a regularization term representing local geometrical structure in the loss function of NMF. I think the idea of this model is interesting and could be useful, but the authors need to better justify the individual components of the model. As for the results, a more informative comparison with other scRNA-seq clustering tools are in need.

Line 24-26: Many clustering analyses will contain a feature selection step, which includes the identification and usage of marker genes. Therefore, it’s not proper to say cell marker genes haven’t been used in the clustering analysis of scRNA-seq data.

Line 100: Is it reasonable to assume the noises in scRNA-seq data are sparse? The authors need to provide evidence for this claim.

Line 103: The authors should introduce all the matrices (e.g., X, W, H) in the context of single cell expression data. Moreover, the matrix L should be briefly discussed here even though it is defined in line 128.
It looks like the authors choose to use the Frobenius form. Why don’t the authors use the L(2,1) norm as discussed in line 92, since it’s shown to be more robust?

Line 107: What does “this property” refer to?

Line 118: The authors need to justify the selection of this heat kernel and the parameters used in the kernel function. Are the results sensitive to the parameters? Will other options possibly lead to better results?
In addition, some cell types only have a few cell markers. The calculated cell distances would be unstable in this case. Did the authors consider identifying de novo cell markers?

Line 155 gives a definition, not a theorem.

Line 160. Why the threshold is alpha/2 in (3)? Should it be 2*alpha?

Line 161: Is this algorithm guaranteed to converge?

Results:

All the datasets tested in the results section have no more than one thousand cells, while large scale single-cell data with thousands of data are very common. The authors need to evaluate the method on larger dataset for a more comprehensive comparison.

There are more than 10 clustering methods that are specifically designed for single cell gene expression data, but the authors didn’t compare rssNMF with any of these methods. It is important to know if rssNMF provide improvement over existing scRNA-seq methods.

Minor:
180 “ a many” looks like a typo

Reviewer 3 ·

Basic reporting

Wu et al presented a new method called rssNMF, which incorporate prior knowledge into the rNMF method. They compared their method to other NMF methods such as rNMF and original NMF, as well as two common clustering methods kMeans and hierarchical clustering, to show rssNMF is superior than all the other methods compared. The language is clear, and reader can follow the logic of the story. The figures and tables are well presented with scripts of the algorithm included.

Experimental design

I have major concerns about the design of the method comparison. Firstly, it is uncommon to define marker gene as the DE genes between different clusters because cluster identity is unknown when performing clustering analysis. Marker genes are usually obtained through previous study, such as bulk RNAseq experiment or fluorescence-activated cell sorting (FACS). For example, one could perform DE analysis of immune cell reference dataset (such as immgen.org) and then use the marker genes obtained on a PBMC dataset for better clustering performance. Secondly, while kMeans and hierarchical clustering are fundamental clustering algorithms, they rarely being used directly to scRNAseq data (Duo et al. F1000Res 2018. Freytag et al. F1000Res 2019), instead they might be a part of a clustering pipeline. As an example, SC3 (Kiselev et al. Nature Methods 2017) use kMeans in their consensus clustering pipeline. Therefore, the rational of choosing these two methods is unclear. The detail of these two methods used is not stated as well, such as the input and the parameters. Authors should either ignore the comparison to other methods thus limiting the scope of the comparison only to different NMF methods or compare against commonly used unsupervised clustering methods such as Seurat and SC3.

Validity of the findings

As mentioned above, the author should provide more detail on the method comparison, such as the parameter used in each method and how exactly the marker genes are selected, such as providing reproducible scripts for DE analysis.

Additional comments

typo:
"hierarch clustering" should be hierarchical clustering
there is formating error in line 201.

---

## Round 0.2 · Major Revisions

The manuscript requires major revision for further consideration. Additionally, the authors successfully implemented SC3 algorithm at the benchmark step however did not properly cite the following articles: PMID:30271584 and PMID:31646845 in their revised manuscript. These studies suggest that SC3 is the "optimum" method among all single cell clustering methods therefore they have particular importance for the field, I believe.

Reviewer 1 ·

Basic reporting

Paper summary:
Authors build a method to incorporate marker genes via matrix Q to better cluster scRNA-seq data. To construct matrix Q authors first use true labels of cells to make groups of cells, then marker genes are selected for each cell group; differentially expressed genes are used as marker genes of each cell group. Then Q is used as input in rssNMF algorithm to obtain optimal H. Matrix H is clustered by KMeans into clusters of cells. These clusters are compared to the "true" labels via ARI. Clustering step is repeated 30 times, average ARI is reported as a performance metric.

Major comments:
(1) Author summary of analysis workflow should be added to the manuscript, focusing on the selection of marker genes.
(2) Tracked Word document is inconsistent with pdf-v1 document. Not all changes that authors claim to be done are incorporated in pdf-v1.
(3) Algorithm 1 missing input parameter L, the key parameter in rssNMF method.
(4) Gene score is determined from matrix W. Gene score is only necessary for selection of feature genes for biological significance validation. What is the cutoff for gene score value to use a gene in enrichment analysis?
(5) On line 300: "As shown in the Fig. 3, we assign all the genes to a specific cluster by different colors. The depth of the color indicates the value of gene score". The caption of Fig. 3 contains matrix W, H and consensus of C, which is not consistent with line 300.
(6) The last dataset ("Zeizel") in Table 2 has ARI of nearly zero, i.e. labels are assigned a random manner. Authors should verify and explain this result.

Experimental design

NA

Validity of the findings

NA

Additional comments

Minor comments:
(1) 57: Subsection title "Related work" is unnecessary.
(2) Any inconsistencies of names and terms need to be corrected; e.g. KMeans, Kmeans, RNMF, rNMF.
(3) Table 1 two bottom lines are not aligned with the rest of the entries.
(4) 215: term "Hierarch clustering" need correction.
(5) A reference for MSigDB is missing in the manuscript.

Reviewer 2 ·

Basic reporting

no comment

Experimental design

no comment

Validity of the findings

no comment

Additional comments

The authors addressed most of my comments, but I still have following concerns after reading the authors’ response and revised manuscript.

Since the definition of weighting matrix is arbitrary, the authors should acknowledge this point in the discussion section and discuss possible improvement or validation approaches.

I didn’t understand this sentence in the authors’ response:
“Line 160: The reason is that the coefficient of Frobenius term in objective function is 1 but 1/2 in the theorem.”

The authors answered my question regarding the convergence but did not discuss this point in the manuscript.

Reviewer 3 ·

Basic reporting

I have no further questions.

Experimental design

I have no further questions.

Validity of the findings

I have no further questions.

---

## Round 0.3 · Minor Revisions

Reviewer #1 recommended some minor changes. For further consideration, I would like invite the authors to revise their manuscript for the last time.

Reviewer 1 ·

Basic reporting

Almost all of my comments have been addressed by the authors in the revised manuscript. One problem has not been addressed and one newly discovered error in the equation needs to be corrected.
(1) The issue of Figure 3 and its caption not being consistent with main text is not addressed. In the latest revised manuscript authors claim that “The depth of the color indicates the value of gene score and we can assign the cluster genes visually.” Matrices W, H, and C are displayed in Figure 3 as noted in the caption, however, the gene score is a function of W and is not shown in Figure 3. Therefore, in Figure 3, the depth of the color does not indicate the value of gene score.
(2) Additionally, the equation on line 64 contains two errors: first, the L denotes a functional and not a graph Laplacian, therefore should be changed to any other name, e.g. f; second, the right-hand side of the equation is missing the min operator, same as used in the left-hand side of the equation.

Experimental design

NA

Validity of the findings

NA

Additional comments

NA

Reviewer 2 ·

Basic reporting

none

Experimental design

none

Validity of the findings

none

Additional comments

The authors have addressed all of my comments.

---

## Round 0.4 · accepted · Accept

I congratulate the authors for their work.